


# Ozone Reactivity Measurement of Biogenic Volatile Organic
# Compound Emissions

Detlev Helmig[1,2*], Alex Guenther[3], Jacques Hueber[1], Ryan Daly[1], Wei Wang[1], Jeong-Hoo Park[1],
Anssi Liikanen[4], Arnaud P. Praplan[4]
[1]Institute of Arctic and Alpine Research, University of Colorado, Boulder, CO 80309, USA
[2] now at: Boulder Atmosphere Innovation Research LLC, Boulder, CO 80305, USA
[3] University of California Irvine, CA, USA
[4]Atmospheric Research Composition, Finnish Meteorological Institute, 00101 Helsinki, Finland
**corresponding author: dh.bouldair@gmail.com**
*Manuscript submitted to*
## Atmospheric Measurement Techniques
October 25, 2021
## Abstract

Previous research on atmospheric chemistry in the forest environment has shown that the total
reactivity by biogenic volatile organic compound (BVOC) emission is not well considered in forest
chemistry models. One possible explanation for this discrepancy is the unawareness and neglect
of reactive biogenic emission that have eluded common monitoring methods. This question mo-
tivated the development of a total ozone reactivity monitor (TORM) for the direct determination
of the reactivity of foliage emissions. Emissions samples drawn from a vegetation branch enclo-
sure experiment are mixed with a known and controlled amount of ozone (e.g. resulting in 100
ppb of ozone) and directed through a temperature-controlled glass flow reactor to allow reactive
biogenic emissions to react with ozone during the approximately 2-minute residence time in the
reactor. The ozone reactivity is determined from the difference in the ozone mole fraction before
and after the reaction vessel. An inherent challenge of the experiment is the influence of chang-
ing water vapor in the sample air on the ozone signal. A commercial UV absorption ozone monitor
was modified to directly determine the ozone differential with one instrument and sample air





was drawn through Nafion dryer membrane tubing. These two modifications significantly re-
duced errors associated with the determination of the reacted ozone compared to determining
the difference from two individual measurements and errors from interferences from water va-
por, resulting in a much improved and sensitive determination of the ozone reactivity. This paper
provides a detailed description of the measurement design, the instrument apparatus, and its
characterization. Examples and results from field deployments demonstrate the applicability and
usefulness of the TORM.


**1. Introduction**

Recent field research on the atmospheric chemistry in forest environments has yielded a

series of results that cannot be explained with our current comprehension of biogenic emissions,
deposition processes, and chemical reactions. These findings date back to the pivotal paper by
*Di Carlo et al.* [2004] that stimulated new interest and research into the question of unaccounted
for biogenic volatile organic compound (BVOC) emissions. These researchers compared the di-
rectly measured hydroxyl radical (OH) reactivity in ambient air at the University of Michigan Bio-
logical Station (UMBS) PROPHET forest research site with the OH reactivity calculated from a
comprehensive set of measured atmospheric gas phase species. The important conclusion of this
study was that identified compounds could only account for about 2/3 of the directly measured
OH reactivity. Interestingly, the difference between the two measurements, often called "missing
OH reactivity" showed temperature dependence very similar to that found for monoterpene
(MT) compounds. This similarity led the authors to hypothesize that the missing OH reactivity is
due to non-identified BVOC emissions emitted from tree foliage at this site.

While these findings were surprising at the time of publication, several other subsequent

studies have come to similar conclusions. OH reactivity measurements in ambient air have con-
sistently shown higher OH reactivity values than what can be accounted for by quantified chem-
ical species, and notably, the review of available measurements shows a tendency towards a



higher discrepancy at sites that are subjected to a relatively high influence from BVOC emissions
[*Lou et al.*, 2010].

The other line of research that has pointed towards the current underestimation of BVOC

emissions relies on ozone flux observation over forest canopies. *Kurpius and Goldstein* [2003]
segregated ozone deposition fluxes over a ponderosa pine plantation into stomatal uptake, non-
stomatal surface deposition, and gas phase chemistry contributions. They found that during sum-
mer, the ozone flux was dominated by gas-phase chemistry, and that the ozone loss showed an
exponential increase with temperature, with similar behavior as BVOC emissions. However, iden-
tified BVOCs could only account for a small fraction of this reactivity. Consequently, these re-
searchers postulated that there is a "large unrecognized source of reactive compounds in for-
ested environments". A follow-up study [*Goldstein et al.*, 2004], based on measurements during
a forest thinning experiment, went even further and claimed that "unmeasured BVOC emissions
are approximately 10 times the measured monoterpene flux". These hypotheses have been sup-
ported by findings from other subsequent studies [*Altimir et al.*, 2004; *Holzinger et al.*, 2005;
*Altimir et al.*, 2006; *Hogg et al.*, 2007; *Fares et al.*, 2010a; *Fares et al.*, 2010b; *Fares et al.*, 2010c;
*Wolfe et al.*, 2011].

There has been considerable progress in identifying and characterizing hitherto unrecog-

nized BVOC emissions. The most significant ones are light-dependent MT emissions [*Ortega et
al.*, 2007; *McKinney et al.*, 2011] and sesquiterpenes (SQT) [*Duhl et al.*, 2008]. Furthermore, it has
been recognized that methyl chavicol can be an important emission [*Bouvier-Brown et al.*, 2009a;
*Bouvier-Brown et al.*, 2009b; *Misztal et al.*, 2010]. However, inclusion of these emissions only
contributes a minor fraction to closing the gap between identified and inferred BVOC emissions.
In a study at the PROPHET site, using the comparative reactivity method, *Kim et al.* [2011] deter-
mined directly the OH reactivity in emission samples drawn from branch enclosures. OH reactivity
was also calculated based on BVOC emissions identified by Proton Transfer Reaction Mass Spec-
trometry (PTR-MS) and Gas Chromatography Mass Spectrometry (GC-MS). A red oak, white pine,
beech, and maple tree were investigated. Their results indicated a high range of total OH reac-
tivity from the emissions of these species, with red oak emissions showing the highest OH reac-



tivity overall. Identified isoprene and MT emissions could explain the directly measured OH reac-
tivity from red oak, white pine, and beech. However, isoprene and monoterpene emissions from
red maple could only explain a fraction of the measured OH reactivity. The OH reactivity from
maple was dominated by emission of the SQT α-farnesene, which is a compound that would not
have been identified in earlier studies of ambient BVOC at this site. These findings show that the
chemical reactivity in emissions from different tree species can vary substantially in their overall
magnitude and attribution to the emitted BVOC species. This indicates that there is the potential
that ecosystems with different plant species composition could have substantial unaccounted for
emissions that contribute to OH reactivity. This suggests that there must be BVOC compounds or
compound classes emitted from foliage that current measurements do not capture, which is not
unexpected given the major analytical challenges associated with analysis of some organic com-
pounds.

In this work, we are describing a monitoring approach that addresses this dilemma by con-

straining the total ozone reactivity of BVOCs emissions with a direct measurement. These obser-
vations can be contrasted with the reactivity that is calculated from the sum of the reactivities of
individual BVOCs and their OH reaction rates to assess the fraction of the identified and missing
compounds that contribute to the total reactivity. The instrument relies on a flow reactor. Sample
air containing BVOCs is mixed with a small flow containing a high mole fraction of ozone. The loss
of ozone is monitored with a differential ozone measurement. Our Total Ozone Reactivity Moni-
tor (TORM) that was previously presented in [*Helmig et al.*, 2010; *Park et al.*, 2013] has since
undergone further testing and development. The calculation of ozone reactivity is explained in
Supplement A, and the modelled decay of a few typically measured BVOC and ozone in the reac-
tor is available in Supplement B.

Two other instruments relying on different types of reactor and detection methodology have

been reported since [*Matsumoto*, 2014; *Sommariva et al.*, 2020]. These previous publications
have also provided the principle and reaction kinetics consideration for this measurement. A lin-
ear double-tube Pyrex glass tube flow reactor with ozone detection up- and downstream of the
reactor by two modified commercial (ECO PHYSICS, CLD770) chemiluminescence detectors (CLD)



465 was used in the work by *Matsumoto* [2014]. The ozone reactivity was determined from the dif-

466 ference of the two analyzers' signal. A 1 m long, 2.4 L volume-PTFE linear reactor was used by

467 *Sommariva et al.* [2020]. These authors used two commercial Thermo Scientific Model 49i UV

468 absorption monitors for the ozone determination, with the ozone reactivity again determined

469 from the difference of the two monitor signals.

470 We particularly emphasize the necessity of properly characterizing the interference from

471 water vapor on the ozone determination, and the advantage of the measurement of the amount

472 of reacted ozone through a differential ozone determination with a single monitor. Thirdly, as-

473 sembly of readily available instrument components facilitate a relatively easy, low expense in-

474 strument assembly.

475 Rigid chambers or flexible bag enclosures are the common approaches for studying biogenic

476 emissions by dynamic or static vegetation enclosures [*Ortega and Helmig*, 2008; *Ortega et al.*,

477 2008]. Enclosure experiments allow the selective identification of emissions from individual plant

478 species. Depending on the operational parameters, emissions can build up to many times, even

479 order of magnitudes, higher levels than in ambient air. Higher temperatures (than in ambient air)

480 are often encountered inside enclosures from the greenhouse warming effect, which enhances

481 emissions and facilitates higher sensitivity of emissions determination. An inherent disadvantage

482 and analytical challenge, however, is the evaporative water flux from the transpiring enclosed

483 foliage. Under the most extreme, and not too uncommon conditions, water vapor saturation can

484 be achieved inside the chamber, causing liquid water condensation on the chamber inside walls

485 and within sampling tubing. The water flux is sensitive to the stomatal conductance, responding

486 to conditions of light and temperature. In an ambient setting, these often change dynamically,

487 causing similarly fast changes in water vapor concentration inside the enclosure and sample air.

488 At 30°C and water saturation, the water vapor mole fraction is approximately 4.2 %. A mere 10

489 % fluctuation equates to 4.2 parts per thousand, or 4,200,000 ppb of a water vapor change. The

490 signals that have been achieved in ozone reactivity monitoring instruments system are usually in

491 the single ppb range. Consequently, for the ozone monitoring to be selective, the ozone detec-

492 tion needs to be insensitive to water vapor changes that can be on the order of $10^6$-$10^7$ times

larger in mole fraction than the ozone signal. This is an enormous challenge for this measure-
ment, as both the ozone CLD and UV absorption measurements are sensitive to water vapor.
Interference with an instrument signal response in the range of tens to hundreds of ppb has
been reported for different types of UV absorption monitors from rapid changes in water vapor
[*Wilson and Birks*, 2006; *Spicer et al.*, 2010]. This interference was traced to humidity effects on
the transmission of light, i.e. reflectivity of light on the cell walls, through the optical cell [*Wilson*
*and Birks*, 2006]. The study identified that the instrument's ozone scrubber amplified this effect,
acting as a water reservoir adding or removing water to the air flow depending on the sample air
moisture content. A 10 % change in the recorded ozone was observed from a 30 to 80 % RH
increase for a UV absorption monitor [*Kim et al.*, 2019; *Kim et al.*, 2020]. Inserting a Nafion dryer
into the sampling path can reduce the water interference, in the best scenario to within equal or
better than ± 2 ppb [*Wilson and Birks*, 2006; *Spicer et al.*, 2010; *Kim et al.*, 2020]. *Sommariva et*
*al.* [2020] found that the ozone wall losses were dependent on the relative humidity in their PTFE
flow reactor.
While CLD analyzers for ozone determination are more expensive to acquire and operate,
they are popular for fast ozone measurements such as for aircraft [*Ridley et al.*, 1992] and eddy
covariance flux measurements [*Lenschow et al.*, 1981, 1982]. Similarly to UV monitors, CLD in-
struments suffer from an interference by water vapor, which in this case is caused by the quench-
ing of the chemiluminescence signal in the reaction chamber [*Matthews et al.*, 1977; *Boylan et*
*al.*, 2014]. A correction factor of 4-5 x 10$^{-3}$ has been proposed, to be multiplied by the water vapor
mole fraction in nmol mol$^{-1}$ [*Boylan et al.*, 2014]. Under moist ambient air conditions, this correc-
tion can account for up to 15 % of the ozone signal. Consequently, following the enclosure system
water vapor estimates above, CLD in an ozone reactivity system may be susceptible to a several
percent interference from changing water vapor, which is on the same order of magnitude as the
observed ozone reactivity observed in the flow chamber system.
Both, *Matsumoto* [2014] and *Sommariva et al.* [2020] used two ozone monitors for determi-
nation of the ozone upstream and downstream of the reactor, with the reacted ozone then de-
termined as the difference of the recordings from both instruments. One objective of this con-



figuration in the *Matsumoto* [2014] work was to achieve a reduction of the quenching interfer-
ence, based on the assumption that both monitors would have similar responses to the water
interferences, with these errors then mostly cancelling out in the differential ozone reactivity
signal calculation. From a measurement and signal perspective, this is a rather disadvantageous
measurement approach for several reasons: (1) the two monitors need to be carefully
synced/calibrated against each other to make sure the instrument offset is characterized and
corrected for so that their readings are consistent; (2) drifts of any of the two monitors, or of
both, will directly transfer to a measurement error in the ozone reactivity signal; and (3), statis-
tically, the calculation of the ozone reactivity will be subject to a relatively large error, as the
ozone reactivity signal is a relatively small value resulting from the difference between two larger
numbers. Any absolute errors in the directly measured values will therefore transfer into a rela-
tively large error of the smaller differential. For these reasons, it would be preferable to measure
the ozone differential through a direct measurement with one monitor. Furthermore, a one mon-
itor measurement would be advantageous in terms of instrument maintenance and cost.
Our experiment presented here overcomes this predicament by modifying a commercial UV
absorption ozone monitor for the direct measurement of the ozone differential. Further, sample
drying was implemented to reduce the aforementioned interference from fluctuations in the
sample water vapor mole fraction. The experiments described here were conducted successively
on two similar systems at the University of Colorado, Boulder, and the Finnish Meteorological
Institute (FMI) in Helsinki, Finland.
## 2. Methods
The basic principle of the ozone reactivity determination of biogenic emissions is illustrated in
Fig. 1. Emissions from vegetation are combined with a flow of ozone-enriched air and allowed to
react in a flow reactor. Ozone is measured upstream and downstream of the reactor with a single
instrument. In the standard configuration of an UV absorption ozone monitor, ozone-containing
air and scrubbed air (ozone-free air) are either measured sequentially (one optical cell) or in

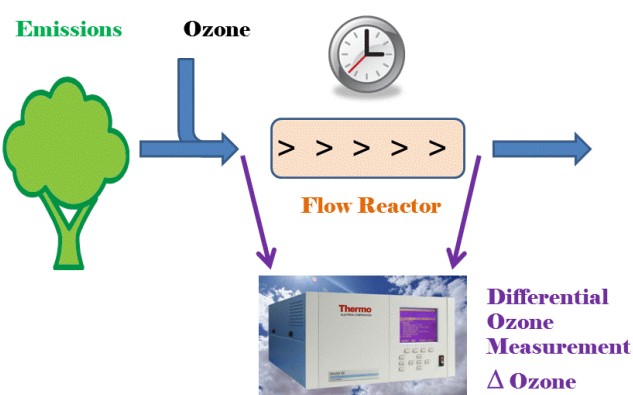

**Figure 1**
Principle of ozone reactivity measurement of biogenic emissions with one monitor that is configured
for differential ozone signal recording.


parallel (two cell instruments), with the ozone mole fraction then determined following the Beer-
Lambert Law. The ozone mole fraction is proportional to the natural logarithm of the light inten-
sity $I$ divided from the sample air (flow 1) by the light intensity in the scrubbed air $I_o$ (flow 2). By
replacing the scrubbed air flow path with a second sampling inlet line, the resulting signal no
longer reflects the difference in ozone between the sample (1) and scrubbed air (2, zero ozone),
but instead becomes the difference in ozone between the two sample flows (2-1). The required
instrument modification is rather simple, illustrated in Fig. 2 for a Thermo Scientific Model 49i
instrument. It requires removal of the ozone scrubber (MoO scrubber in most cases) and the
separation of the scrubbed and sample air into two separate inlets. In the standard configuration,
the 49i samples air at $\approx$ 1.2 L min$^{-1}$ through one inlet. In the modified configuration, this flow is
split in half to $\approx$ 0.6 L min$^{-1}$ each for the Sample 1 and Sample 2 inlets. An early configuration of
the experiment to illustrate how the differential ozone monitoring was evaluated against the
monitoring of ozone up and downstream of the reactor with two instruments is presented in





**(A) Original Pluming Configuration**

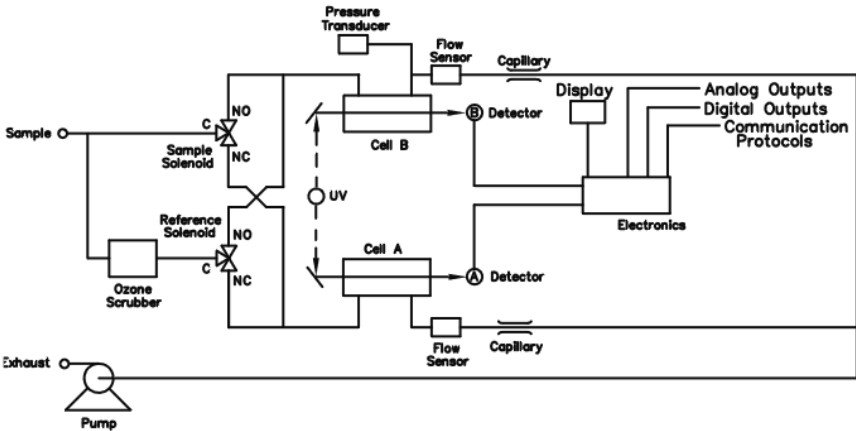

**(B) Differential Ozone Monitoring Pluming Configuration**

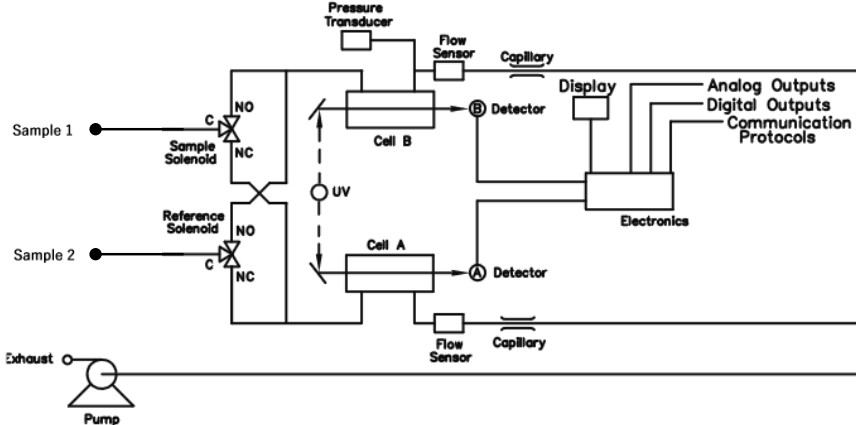

**Figure 2**
Plumbing configuration of a Thermo Scientific Instruments model 49 ozone UV absorption monitor in its original configuration (top) and in the modified configuration (bottom) for monitoring of ozone differentials.




Supplement C; the final one-monitor TORM configuration is shown in Fig. 3. The direct differential
ozone measurement was always conducted with a Thermo Scientific Model 49i monitor. During
the evaluation experiments, several different UV absorption ozone monitors were used for com-
paring the direct measurement with a result from two individual instruments. Those included
Thermo Scientific Model 49i, Model 49C, and a MonitorLabs model 8810 monitor.
While other studies [*Matsumoto*, 2014; *Sommariva et al.*, 2020] utilized linear flow reactors,
this experiment relied on using four glass flasks that were plumbed in series. The glass flask re-
actor design was chosen because it was deemed more compact and robust for field deployment
applications. The 2.5 L borosilicate flasks that were used are air sampling flasks that are routinely
deployed in the NOAA Cooperate Sampling Network for the global sampling of greenhouse gases.
These glass flasks have been developed and extensively tested for their inertness and purity to-
wards atmospheric trace gases (https://www.esrl.noaa.gov/gmd/ccgg/flask.html; flasks are fab-
ricated by Allen Scientific, Boulder, CO). Flasks are covered with a protective film and have two
ports with stopcock Teflon vales. One valve connects to a dip tube that leads to the inside on the
opposite side of the flask (Fig. 4). This configuration allows efficient purging and replacement of
the air volume inside the flasks with minimal mixing. The flasks were plumbed such that the in-
flowing air was always introduced through the dip tube. The four flasks in series add up to a total
≈10 L reactor volume. The flasks are contained in an 45 cm x 45 cm x 45 cm (inside dimension)
Pelican model 0340 cube case (Torrance, CA) that was fitted with 5 cm foam insulation on the
inside. A rope heater, temperature probe, and temperature controller allow to thermostatically
control the temperature, typically to 40°C. The ozone reactant gas was provided from the Thermo
Scientific 49i monitor using its integrated ozone generator. The output was set to provide a 1000
ppb constant output, so that the 1:10 dilution with the sample air flow resulted in a 100 ppb
ozone mole fraction entering the reactor. All experiments described in this paper were conducted
at this 100 ppb ozone mole fraction, unless stated otherwise. A mixer made of Teflon material
(7.50 mm OD with 30 mixing elements, Stamixco AG, Wollerau, Switzerland) was inserted up-
stream of the introduction of the ozone gas flow for providing turbulent mixing between the
sample air and ozone-enriched air. All tubing was made of 6.4 mm o.d./4.7 mm i.d. PFA tubing.




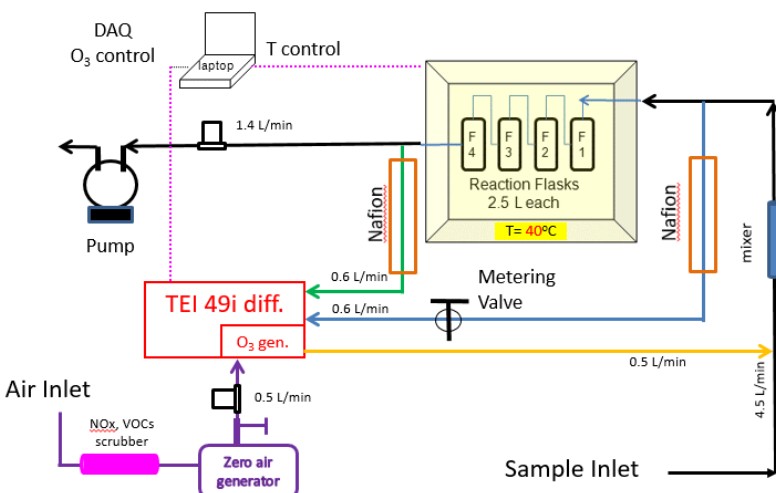

**Figure 3**
Final configuration of the total ozone reactivity monitor (TORM) using one Thermo Scientific (TEI) 49i PS monitor plumbed for the direct differential ozone measurement (Figure 2), and with the Nafion dryers and metering valve included. Flow rates are indicated in the figure. Total flow through the reactor is 5 L min$^{-1}$.


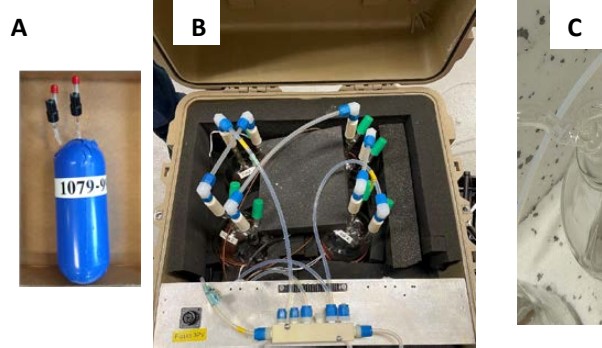
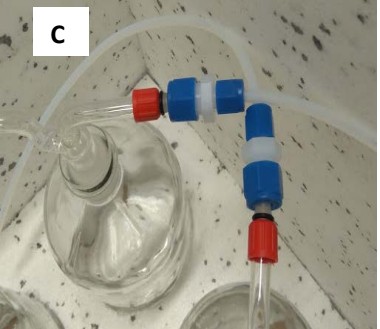

**Figure 4**
(A) Photograph of one of the glass flasks that were used for the University of Colorado, Boulder flow reactor. (B) The ozone reactor with four of the flasks plumbed in series contained in an insulated and temperature-controlled field-deployable enclosure. Four flasks were plumbed in series for a total flow reactor volume of 10 L.  (C) The 2-L bottles (borosilicate glass 3.3) used in the Finnish flow reactor system.




Experiments did not consider adding an OH scavenger (i.e. cyclohexane) [*Matsumoto*, 2014;
*Sommariva et al.*, 2020]. *Sommariva et al.* [2020] estimated a < 6 % difference in ozone reactiv-
ity for BVOC ozonolysis reactions based on modeling, but could not identify differences with
and without cyclohexane added in their experiments. It is therefore unlikely that addition of an
ozone scrubber will make a notable difference in the ozone reactivity monitoring results. The
instrument operation and signal acquisition were controlled via a National Instruments digital
input interface and custom-written LabView software.

During field deployments, branch enclosures were set up on sweetgum (Liquidambar

styraciflua L.), white oak (Quercus alba), and loblolly pine (Pinus taeda) tree branches following
our previously described protocol [*Ortega and Helmig*, 2008]. A Tedlar bag (36"x24") was
wrapped around a tree branch of the size that when the bag is inflated, the branch was situated
in the middle of the bag with minimum touching of the wall. Scrubbed ambient air free of $NO_x$
ozone and BVOC (Purafil and activated charcoal scrubbers),was delivered to the enclosure at 25
L min$^{-1}$. Most of the moisture in the purge air was also removed by passing it through a set of
coils placed inside a refrigerator. The scrubber system did not remove carbon dioxide. Air sam-
ples from the enclosure were taken through the ports affixed on the Tedlar bag, drawn at flow
rates that are suitable for the sampling apparatus and instruments. The rest of the purge air es-
caped the enclosure mainly through the gap between the bag and the main stem of the branch.


**3. Results and Discussion**

**3.1 System conditioning**

A newly assembled system exhibited a significant ozone sink, on the order of 20-30 ppb loss

of ozone (at 100 ppb) at a 5 L min$^{-1}$ reactor flow. The slow decline of the ozone loss signal over
time indicated a gradual equilibration of the system to the ozone in the sample air. This ozone
loss and signal drift could almost entirely be eliminated thorough conditioning of all tubing and
the reactor with an air flow enriched in ozone. For this conditioning, the system was purged for
24 hours with 500 ppb of ozone. After this treatment, the ozone loss associated with the sample
flow through the reactor in the absence of chemical gas reactants, i.e. the reactor background
signal, was, depending on the particular system condition and operational variables, on the order
of 1-2 % of the supplied ozone mole fractions; i.e. at 100 ppb ozone, the loss was reduced to 1-2
ppb and did no longer show any drifts in the signal. After warmup, the 1-min averaged $\Delta[O_3]$
signal displayed a standard deviation ($\sigma$) of 0.075 - 0.096 ppb (over 1 h, n = 60). This translates
into a limit of detection ($3\sigma$) of $1.8 - 2.3 \times 10^{-5}\,s^{-1}$ for the reactivity (for a theoretical residence
time of 150 s, and correcting for the ozone dilution flow). This sensitivity is slightly higher, i.e.
resulting in a lower limit of detection than that reported by [*Matsumoto*, 2014] ($4 \times 10^{-5}\,s^{-1}$, for a
residence time of 57 s), and approximately 2-3 times lower than that reported by [*Sommariva et*
*al.*, 2020] ($4.5 - 9 \times 10^{-5}\,s^{-1}$ for a residence time of 140 s). The stability of the ozone reactivity signal
was tested on the Finnish system over a full day, with the reactor located outside and sampling
from an empty enclosure that was subjected to a full daily cycle of changing ambient conditions
in temperature, humidity, and light. There was no notable drift in the $\Delta[O_3]$ signal over the meas-
urement period despite the changes in the environmental conditions (Supplement D).

**3.2 Balancing of the ozone monitor inlet pressures**
The readings from the differential ozone monitor are sensitive to the difference in the pres-
sure in the two sampling lines that connect to upstream and downstream of the reactor (Supple-
ment E). The pressure differential results from the vacuum generated by the sampling pump for
providing flow through the reactor. The 49i diagnostics menu allows monitoring of the pressures
of the two optical cells. In the original configuration, it was found that there was a pressure dif-
ference of, depending of the flow rate, 20-30 torr between the two cells at a $5\,L\,min^{-1}$ reactor
flow, with the lower pressure recorded in the line downstream of the reactor. This pressure dif-
ferential alters between negative and positive values as the monitor alternates air from the two
inlets through the two optical cells. This pressure difference results in an artificial ozone signal
offset between the two sampling paths. An increase of the flow rate through the reactor causes
a change in the pressure difference and the ozone differential reported by the monitor: Increas-
ing the flow rate from 2 to $9\,L\,min^{-1}$ corresponded to an increase from 2 to 7 ppb increase in the



differential ozone signal. This behavior is clearly a measurement artifact and counter to the ex-
pected ozone loss, as the actual chemical ozone loss decreases with decreasing residence time
of the air inside the reactor (i.e. increasing flow rate). This measurement artifact was mitigated
by inserting a 0.64 cm Teflon metering valve into the sampling line upstream of the reactor. By
closing the valve slightly, the flow was restricted to where both cell pressure readings from the
reactor were equal (within ≈1 torr). This resulted in an ozone differential signal of ≈1.7 ppb that
was insensitive to the reactor flow rate (Supplement E). The final plumbing configuration of the
TORM and its integration into a vegetation enclosure experiment is shown in Fig. 5.

**3.3 Evaluation of the direct differential ozone reactivity measurement**
Results from the parallel operation of two ozone monitors measuring the actual ozone be-
fore and after the reactor, with Δ[O3] calculated from the difference of the two readings, com-
pared to the direct ozone differential measurement by TORM are summarized in Fig. 6. Field data,
collected during the Southern Oxidant and Aerosol Study (SOAS) (CU Boulder system), constitute
a total of ten days of measurements collected using branch enclosures on three different
branches of sweetgum trees. The ozone differential was normalized to the air flow through the
chamber and to the dried weight of leaf biomass that was sampled from the vegetation in the
branch enclosure. These time series data show a clear diurnal cycle with the ozone reactivity
increasing steeply during daytime hours. Results are reasonably consistent between days and the
three different enclosures, considering that the BVOCs emissions that determine this signal are
highly sensitive to light and the enclosure temperature, which varied during the experiment.
There is high agreement between the ozone reactivity results from both configurations across
these experiments. A linear regression between results from the two monitoring methods from
the SOAS study yields a slope value of 0.996. The graphed data also show the substantial im-
provement in the noise of the measurement with the direct differential monitoring (A, B). The
precision error of the direct differential measurement is only about 1/5 compared to the result
from the two monitors. After the system equilibration, the 1-σ standard deviation of the differ-
ential ozone measurement for 1-min averaged readings was generally in the range of

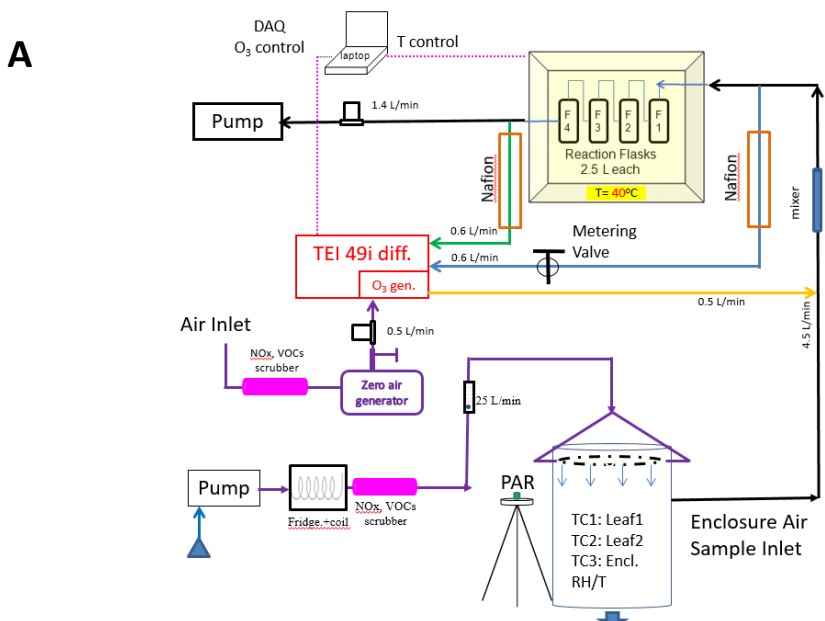

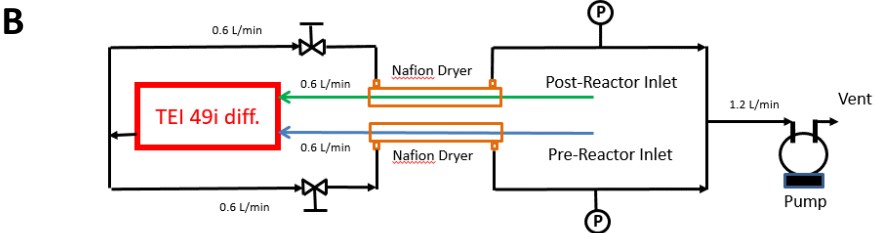

**Figure 5**
(A) Final configuration of the total ozone reactivity monitor with one differential ozone monitor, the sampling line pressure balancing valve, and the Nafion dryers. Schematic (B) shows the detail of the Nafion Dryer plumbing including the external pump that was added to the system for providing the purge flow for the Nafion dryers.




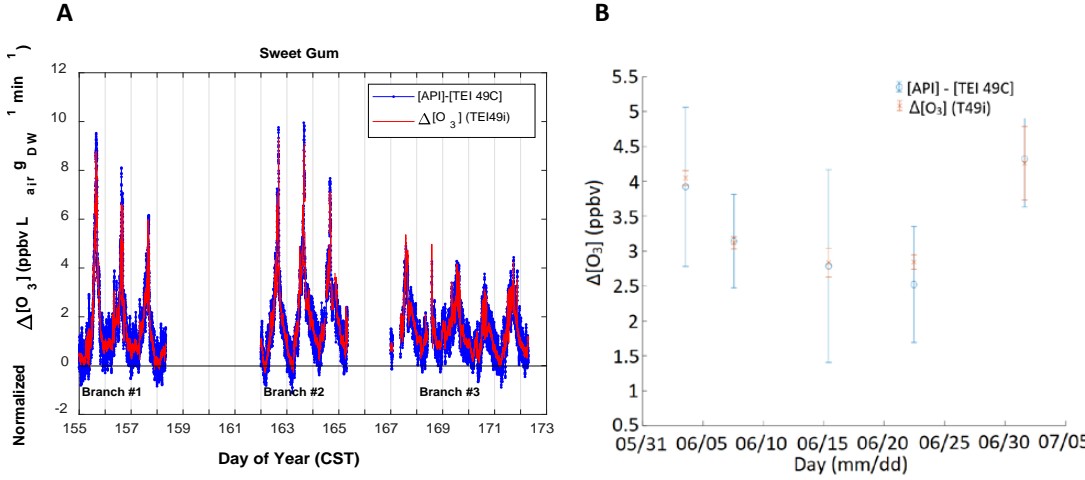

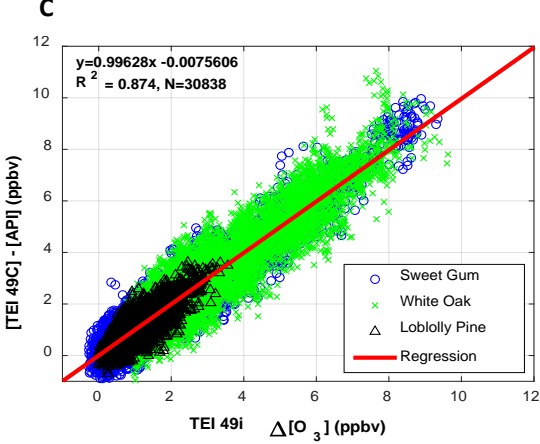

**Figure 6**

Results from comparisons of monitoring the ozone loss in the reactor with two monitors versus measuring the ozone differential directly with the configuration shown in Figure 2B. (A) Three multi-day experiments of ozone reactivity monitoring from an enclosure of sweetgum branches. (B) $\Delta[O_3]$ determinations from blank experiments on an empty enclosure. (C) Summary results of experiments on a total of three different vegetation species. All field experiment results are from the Southern Oxidant and Aerosol Study (SOAS) campaign between June to July 2013 at a field site in Perry County, west central Alabama (Praplan et al., in preparation).


0.1 – 0.2 ppb, which was 2-3 times lower than the calculated ozone difference from the two-

monitor measurement. These results clearly indicate the benefits of the single monitor measure-
ment: (1) the accuracy of the ozone reactivity measurement is consistent with the differential
two-monitor determination; (2) there is a very significant improvement in the measurement pre-
cision from using a single monitor; and (3) the operation of a single monitor is less tedious and
labor intensive as it does not require the regular intercomparison for determination of offsets





and drifts and correction algorithms for calibrating the response of two individual monitors
[*Bocquet et al.*, 2011; *Sommariva et al.*, 2020].
**3.4 Sample residence time in the reactor**
The desired operation of a flow reactor system is for air to move through the reactor as a
narrow plug, with minimal turbulence and mixing. Most flow reactors are tubular and linear and
are used in laboratory settings. Depending on their operational variables, they achieve seconds
to a few minutes residence time. The residence time and peak broadening during transport
through the reactor was studied by installing a syringe injection port upstream of the reactor,
injection of a small volume of a 1 ppm standard of nitric oxide (NO), and monitoring the ozone
loss from the ozone + NO reaction downstream of the reactor with a fast-response (5 Hz) nitric
oxide chemiluminescence instrument. Experiments were conducted in two different configura-
tions:  1. In the normal plumbing configuration, with the incoming air introduced to each flask
through the dip tube.  2. To test the effect of the dip tube, the plumbing was also reversed. The
flow through the reactor was set to 4 L min$^{-1}$, which for an ideal flow reactor, at 10 L volume,
should result in a 2.4 min (150 s) residence time. Results of these tests are shown in Fig. 7. For
both configurations, the peak signal was observed earlier than the theoretical time, i.e. ≈30 s for
the normal configuration, and ≈50 s for the reversed configuration. The peak widths (at half of
peak maximum) were ≈90 s and 120 s, for the normal and reversed configuration, respectively.
The behavior in these data show that there is a considerable amount of mixing inside the reactor
glass flasks, causing deviation from an ideal flow reactor. Nonetheless, the residence time of ≈120
s for the normal plumbing configuration is sufficient to meet the requirements for the ozone
reaction experiment. The findings from this experiment were confirmed at a higher, 6 L min$^{-1}$
flow rate (Supplement F). Both experiments show the advantage of the air introduction through
the dip tube, resulting in a narrower peak, i.e. narrower defined residence time.

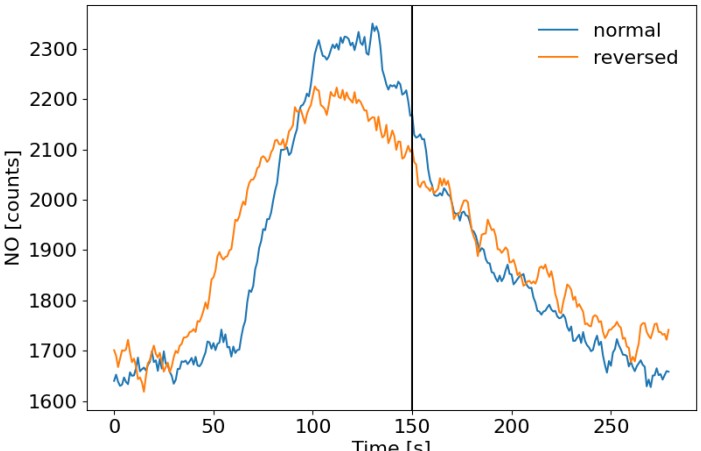

**Figure 7**

Test of sample air residence time in the flow reactor. A small volume of a 1 ppm NO standard was injected through a port upstream of the reactor and NO was monitored downstream with a fast response chemiluminescence analyzer (1 s time resolution). 5 s running averages are presented here. The normal configuration was with the flow entering each flask through the dip tube. The reversed configuration was with the air low exiting each flask through the dip tube. The vertical black line indicates the theoretical residence time based on the total flow rate (4 L min⁻¹) and total volume (10 L) of the reactor, assuming that there was no mixing inside the flasks.

**3.5 Evaluation and Mitigation of Humidity effects**

As elucidated on in the introduction section, changes in humidity can severely interfere in

the ozone determination. Ozone monitors have been found to be less sensitive, i.e. report ozone
below its actual value at high humidity, and to exhibit large artificial signal fluctuations from rapid
changes in the sample water vapor. Characterization and mediation of the sensitivity of the ozone
reactivity measurement to water vapor was a main emphasis of our experiments. Earlier experi-
ments, where the sampling flow was subjected to variable water vapor, such as by injecting small
volumes of water through an injection port upstream of the reactor in the configuration shown
in Supplement C, confirmed the findings from prior literature: Despite a constant ozone mole
fraction that was fed into the reactor, both, the two-monitor determination, and the single mon-
itor ozone differential determination, showed instantaneous changes in the ozone signal, reach-
ing on the order of 10 ppb. The bias in the ozone recording lasted significantly longer (≈10 times)



than the residence time that was determined in the above described experiment using nitric ox-
ide. These water vapor effects on the ozone signal were mitigated by two modifications to the
TORM: (1) the glass flasks reactor was insulated and a heater, regulated by a temperature con-
troller was added to control the temperature of the reactor to 40°C. This heating significantly
reduced the residence and interference time from the water injection, likely due to a reduction
of the adherence of the water vapor to the walls of the glass flasks and other reactor compo-
nents. Our observations agree with the findings reported by *Wilson and Birks* [2006], who found
a reduction of the water interference for their 2B Technologies ozone monitor when the glass
optical cell was slightly heated; and (2) Nafion dryers (0.64 cm o.d. x 180 cm length; MD-110-72
gas dryer, Perma Pure LLC, New Jersey, USA) were inserted into both ozone monitor inlet flows
before and after the reactor. We installed the two Nafion dryers there, rather than one Nafion
dryer for the sample flow path going into the reactor, to prevent possible losses of polar and
unsaturated compounds from the sample flow passing through a Nafon dryer, as has been re-
ported in other prior research. The purge flow for the Nafion dryers was provided by the vent
flow from the TEI 49i. The analyzer vent flow was split into two approximately equal fractions,
resulting in 0.6 L min$^{-1}$ flow for each Nafion Dryer (Figure 5B). Throttle valves were installed in
both lines as flow restrictors and adjusted such that the pressure in the exterior chamber of the
Nafion dryers was ≈10 % below the interior section of the dryer (cell pressure readings from the
differential 49i monitor). The Nafion dryers were conditioned using the same protocol as for the
reactor (see above), after which there was no notable ozone loss from sampling the ozone-en-
riched air flow through the Nafion tubing, in agreement with other previous studies that have
reported negligible ozone loss in Nafion tubing materials [*Wilson and Birks*, 2006; *Boylan et al.*,
2014; *Kim et al.*, 2020].
Results from an experiment with the Nafion dryers in use and where water vapor was in-
creased in multiple steps is shown in Fig. 8. The same humidification system as described by
*Boylan et al.* [2014] was used for moisturizing a zero air dilution gas fed to the TORM. The result-
ing humidity was recorded with a LICOR model 7000 $CO_2/H_2O$ gas analyzer downstream of the
mixer, but upstream of the reactor. Each humidity level was maintained for 30 min, before sub-
jecting the system to the next higher moisture level by a rapid change in the humidity generator




setpoint. The ozone reactivity signal was monitored with the differential 49i monitor, as well as
by recording the absolute ozone upstream and downstream of the reactor with two individual
monitors. Both ozone monitoring systems were sampling through the Nafion tubing. Results of
the experiment (Fig. 8) show a residual ozone reactivity signal response of ≈0.5 ppb over an ≈10
to 84 % RH span for the differential monitor. The two-monitor $\Delta[O_3]$ response is approximately
six times as large. The spikes seen during the moisture transition periods seen in earlier experi-
ments disappeared completely for the differential monitor.

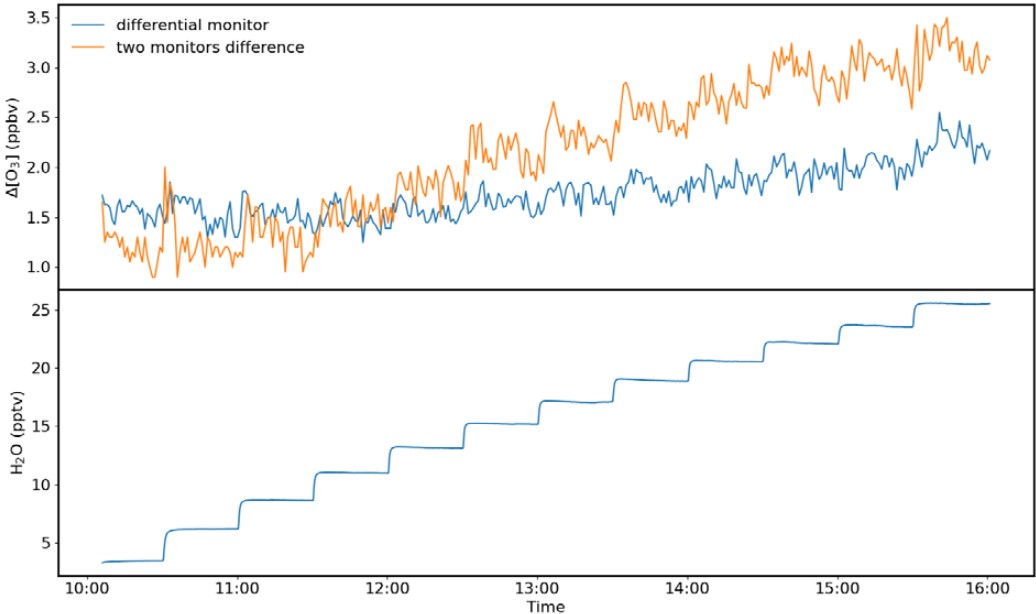

**Figure 8**
Experiment with increasing humidity in the air supplied to the TORM. The humidity content of the sample air is displayed in the lower graph in units of parts per thousand (ppt). A total of 12 levels were adminis-tered, from ≈3 -26 ppt, which at room temperature conditions (25°C) is approximately equivalent to a RH range of 10-84 %.


Similar order of magnitude results were obtained in a series of experiments where liquid
water (20 to 100 μl) was injected into the sampling flow through a septum port upstream of the
reactor. The Nafion dryer removed ≈2/3 of the water interference, and the differential monitor
response to the water injection was less than half compared to calculated difference from the
two-monitors configuration (Supplement G).





**3.6 Application Examples**
Ozone reactivity of test mixtures and samples from vegetation enclosures were investigated
in laboratory and field systems. A laboratory experiment using a flow of limonene standard is
presented in Fig. 9. The gas standard was prepared in house for a target mole fraction of 20 ppm.
However, the actual mole fraction is expected to have decreased with time, but could not be
independently verified at the time of the experiment. The reported mole fractions, after mixing
of the standard with the dilution flow, range between 0 -33 ppbv, which is a typical range

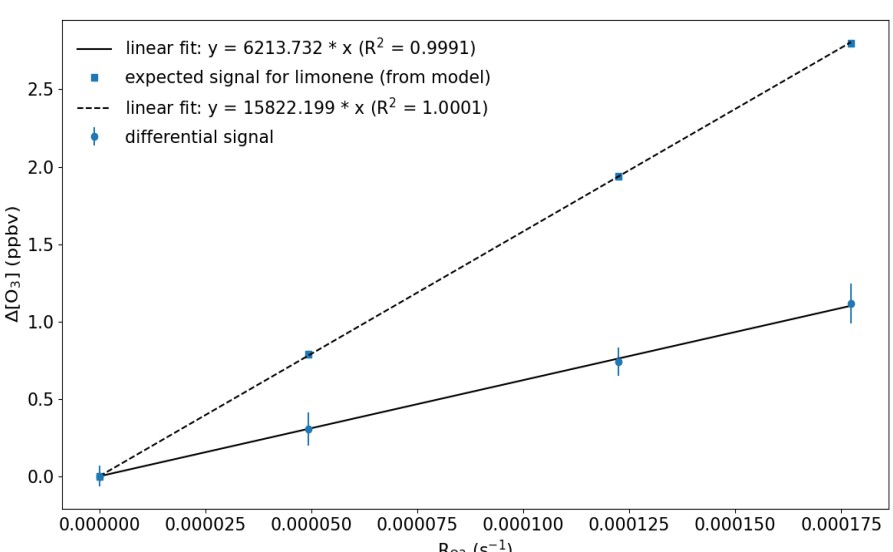

**Figure 9**
Laboratory test of the TORM. A small flow of a high mole fraction limonene standard was fed into
the system upstream of the reactor. The theoretical reactivity calculated from the BVOC ozone rate
constant, ozone mole fraction, and residence time are given on the x-axis.  Error bars represent the
standard deviation for the monitoring data at each level.

observed during enclosure experiments) and represents upper limit values for the mole fraction.
The TORM determination shows good linearity, with a $R^2$ result of the linear regression of 0.9991.
At the highest limonene level, the TORM signal, recorded with the differential ozone monitor,
was 0.9 ppb (after subtraction of the 1.7 ppb $\Delta$ ozone reactor background that was determined
for this particular application).



In Fig. 9, the experimental results from the limonene experiments are also compared with
the modeled signal for various $O_3$ reactivity values for limonene for the operating conditions of
TORM during this experiment. The modeled results reflect the expected $O_3$ decrease due to the
reaction with limonene after the reaction corresponding to the theoretical residence time in the
reactor (here 167 s; 3.6 l min$^{-1}$ flow through a 10 L reactor). The applied rate constant for the
reaction of ozone with limonene at 298 K is 21 x 10$^{-17}$ cm$^3$ s$^{-1}$ [*Atkinson and Arey*, 2003]. A linear
regression shows that Δ[$O_3$] is linearly dependent with RO$_3$ (ca. 1.5 ppbv/(10$^{-4}$ s$^{-1}$)). The discrep-
ancy between the model and the experiment stem likely from the uncertainty of the mixing ratio
in the limonene standard. The experimentally determined sensitivity, i.e. approximately 0.5
ppbv/(10$^{-4}$ s$^{-1}$), is therefore a lower limit. Applying a lower limonene mole fraction in the standard
would lead to a proportionally higher value.
The TORM has been deployed in field settings at several research sites in the U.S. and in
Finland. Fig. 10 displays results from one of these field experiments, i.e. a 3-day branch enclosure
experiment on a red oak tree at the University of Michigan Biological Station. These data show
results from the 2$^{nd}$ and 3$^{rd}$ days of the experiment. The experiment was conducted on relatively
warm and sunny days as can be seen in the radiation and temperature data. Besides the ozone
reactivity signal, shown in panel A, the figure also includes the concurrent measurements of pho-
tochemical active radiation (PAR) (B), respiration and photosynthesis (C), and leaf and enclosure
temperature (D). The change in humidity, reaching a maximum of on the order of 25 parts per
thousand as the mid-day maximum when foliage respiration peaks, confirms our estimate pre-
sented in the introduction section for the humidity changes during vegetation enclosure experi-
ments. Emission samples collected from this enclosure and analyzed by gas-chromatography
showed that emissions from this branch were dominated by isoprene, with further substantial
emissions of MT and SQT compounds. On both days, the TORM recorded a mid-day maximum
differential ozone signal of 12-14 ppb, dropping to 2-3 ppb at night. The instrument readings are
quite similar on both days. The ozone reactivity clearly follows a daily cycle, with low values dur-
ing nighttime hours, and daytime maxima during the early afternoon. The ozone reactivity signal

**Figure 10**
Results obtained over two days from a branch enclosure experiment on a red oak tree, with data for the ozone reactivity measurement (A), solar radiation (B), respiration and photosynthesis expressed as the difference in the water and $CO_2$ mole fractions in the  air stream going into and out of the enclosure (C), and leaf, inside enclosure, and ambient temperature (D).



817 maxima coincide with the peak in diurnal radiation, respiration, and photosynthesis, which sug-

818 gests that the ozone-reactive emissions are modulated by light availability. Comparison of the

819 observed ozone reactivity with the calculated ozone reactivity from identified BVOC species could

820 only account for a fraction of the observed reactivity (*Praplan et al., manuscript in preparation*).

821 Similar diurnal cycles of ozone reactivity were observed for sweetgum in the Southern Oxidant

822 and Aerosol Study [*Park et al.*, 2013], as can be seen in the ten days of measurements shown in

823 Fig. 5. Please note that the data in Fig. 5 were normalized to the leaf dry mass of the enclosure

824 foliage.

825  A presentation of the ozone reactivity results normalized to the leaf dry mass and as a func-

826 tion of leaf temperature for experiments performed at UMBS is shown in Fig. 11. All four species

827 show an increase of reactivity with increasing temperature. This feature indicates that all species

828 emit reactive volatiles at increasing rates as temperature increases. Interestingly, the normalized

829 reactivity is quite different, varying by at least a factor of three. It also appears that the temper-

830 ature dependencies are different, with red maple showing a more dynamic increase than other

831 species. Remarkably, white pine, a high MT emitter, gave the lowest reactivity results. Further-

832 more, red maple results appear to be higher than for red oak, despite the fact that red oak was

833 found to emit high amounts of BVOC, totaling ≈100 x those of maple, but with most of the emis-

834 sions made up by isoprene. The relatively high levels of ozone reactivity are also noteworthy in

835 light of the independent OH reactivity study by *Kim et al.* [2011], who found that red maple emis-

836 sions exhibited the highest missing OH reactivity associated with SQT in comparison with these

837 other three species. Consequently, red maple is a prime candidate for having reactive BVOC emis-

838 sions that hitherto have not been chemically identified.

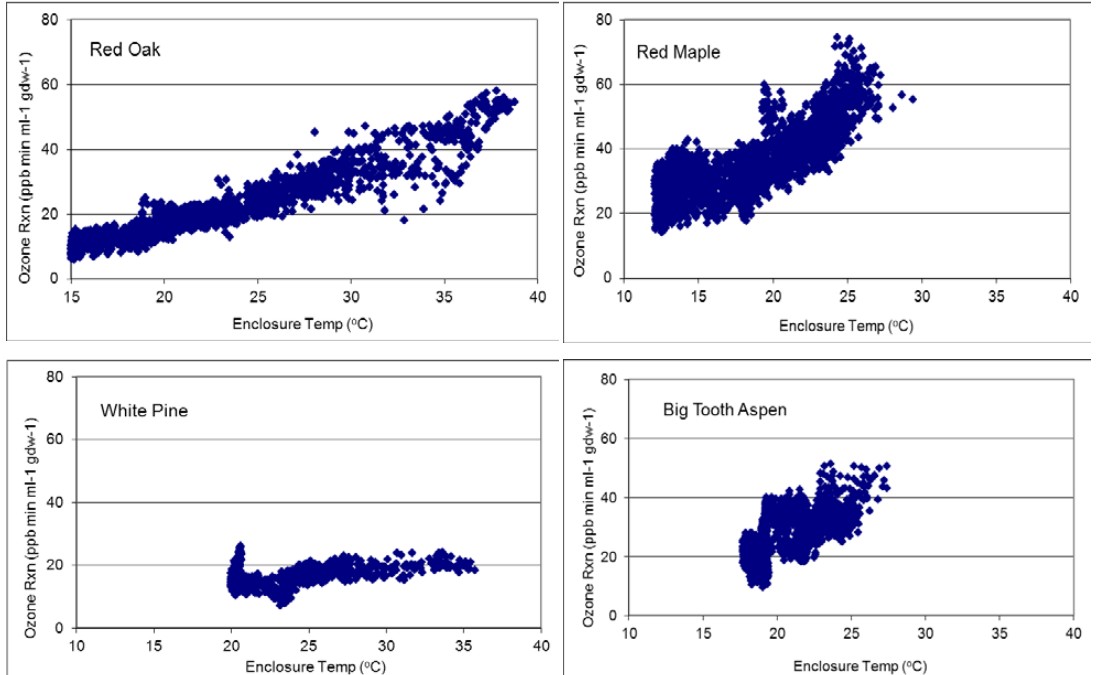

**Figure 11**
Ozone reactivity results from experiments on red oak, red maple, white pine, and big tooth aspen, normalized to the amount of leaf dry mass and flow rate, as a function of enclosure temperature.


## 4. Summary and Conclusions


A total ozone reactivity monitor, TORM, was developed for the study of the ozone reactivity
of biogenic emissions. TORM builds on standard laboratory equipment and can be assembled
with moderate technically skilled personnel and at relatively moderate cost. The instrument was
thoroughly characterized, and a number of ameliorations were implemented that significantly
improved the measurement sensitivity and reduced the interference from absolute and changing
water vapor in the sample air. Critical improvements over previously reported measurement ap-
proaches were the adaptation of a commercial ozone UV absorption monitor for direct measure-
ment of the reacted ozone (ozone differential), heating and temperature control of the reactor,
and the drying of the sample flows with Nafion dryers. Specific challenges arose with this setup



that could be overcome, such as balancing the pressure difference for each cell in the differential
ozone monitor (one cell measuring before the reactor and the other cell measuring after).
TORM has been used in a number of field settings and proven the feasibility and value of this
new measurement. Ozone reactivity signals on the order of 5-0 ppb have been obtained in en-
closure experiments on high-BVOC emitting species. These signals are 20-50 times above the
noise level of the measurement. Chemical identification of BVOC emissions from the enclosure
and estimation of the total reactivity of identified emissions has been able to only account for a
fraction of the directly measured ozone reactivity. Detailed description of these field studies and
discussion of the results, including the attribution of the directly measured ozone reactivity to
identified BVOC emissions, will be presented in a forthcoming publication (*Praplan et al., in prep-*
*aration*).
**Data availability**
All data that the work builds on are presented in the manuscript and Supplemental Information.
**Disclaimer**
This study does not necessarily reflect the views of the funding agencies, and no official endorse-
ments should be inferred.

**Funding Information**
The development and testing of the TORM system has been made possible through funding from
the U.S. National Science Foundation, grants #AGS 0904139, ATM-1140571, and AGS-1561755,
as well as funding from the Academy of Finland (decisions nos. 307797 and 314099).

**Author contribution**
D.H. Principal Investigator of the U.S. study, managed research grants, oversaw the study, pre-
pared and approved the manuscript.
A.G. Co-Principal Investigator of the U.S. study, reviewed and approved the manuscript.
J.H. Constructed instrumentation and conducted experiments, developed control and data ac-
quisition software, approved the manuscript.
R.D. Constructed instrumentation and conducted experiments, participated in field studies, re-
viewed and approved the manuscript.



W.W.  Constructed instrumentation, conducted experiments, prepared, reviewed, and approved the manuscript.

J.H.P. Constructed instrumentation, developed instrument control software, conducted lab and field experiments, reviewed and approved the manuscript.

A.L. Constructed instrumentation, conducted lab and field experiments, approved the manuscript.

A.P.P. Principal Investigator of the Finnish study, conducted field and lab experiments, prepared and approved the manuscript.

**Competing Interests**

The authors declare that they have no conflict of interest.

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
