# Peer review of "Ozone Reactivity Measurement of Biogenic Volatile Organic"

_Atmospheric Measurement Techniques, 2021_

## Author Comment (AC1)

**Ozone Reactivity Measurement of Biogenic Volatile Organic Compound Emissions**

Detlev Helmig[1,2]*, Alex Guenther[3], Jacques Hueber[1], Ryan Daly[1] , Wei Wang[1] , Jeong-Hoo Park[1], Anssi Liikanen[4], Arnaud P. Praplan[4]

[1]Institute of Arctic and Alpine Research, University of Colorado, Boulder, CO 80309, USA
[2]now at: Boulder Atmosphere Innovation Research LLC, Boulder, CO 80305, USA
[3]University of California Irvine, CA, USA
[4]Atmospheric Research Composition, Finnish Meteorological Institute, 00101 Helsinki, Finland
**\*corresponding author: dh.bouldair@gmail.com**

*Manuscript submitted to*

**Atmospheric Measurement Techniques**

**Response to Anonymous Referee #1**

The manuscript "Ozone Reactivity Measurement of Biogenic Volatile Organic Compound Emissions" by Helmig et al. presents a prototype instrument for the direct measurement of total ozone reactivity. This type of instrument has been proposed before, but the authors describe a different design with potentially better performance. Although the subject of the manuscript is clearly within the scope of AMT, I find that there is a general lack of details and information. Several of the experiments are not well described, and in many cases the reader is left to interpret the figures and diagrams to understand what was done and why. Moreover, there are several inconsistencies and errors in the text (e.g. about the residence time in the reactor and the calculation of the ozone reactivity) and some statements are not supported by the data as presented. A model is mentioned at various points, but is never described (not even with a reference to another publication). I would recommend that the authors thoroughly revise the manuscript and resubmit it.

***We thank anonymous referee #1 for their assessment of our manuscript. Considering their comments, we have revised the manuscript to address their concerns, especially regarding the lack of information, the inconsistencies, and errors highlighted in this review. We added more description to the discussion of the residence time and the calculation of the ozone reactivity, and we provide more explanation of the model. Following are the detailed answers to the reviewer's comments.***

MAIN COMMENTS

The Introduction and Methods sections are very long. I would consider dividing them into subsections so that the material is organized better and easier to read. Two instruments appear to be described (one from CU and the other from Finland), but it is not clear whether they are

identical (or what are their differences) and how they were used/deployed during this work. I assume not all the experiments described in the paper were done with both instruments at the same time. Ambient measurements in Finland are mentioned at various places in the manuscript, but the only data shown appear to be from Michigan (USA).

*We added new text that explains in more detail the evolution of the experimental systems and the collaborative work between the U.S. and Finnish groups. We also clarified which experimental results are from which instrument. We consider the international U.S. – Finnish collaboration a strength of this work. This is now emphasized more in the revised manuscript text. Students and postdoctoral scientists from both countries participated in this research. The Finnish group visited the CU Boulder group for a full month to get trained in the instrument design and its operation. Several of the described experiments were conducted during this academic exchange. A joint field campaign was conducted two years later at the University of Alaska Toolik Field Station. Experimental results from the CU and Helsinki instruments were compared on several occasions. The parallel development and comparison of results from the two systems add confidence in the instrument performance and reproducibility of the measurement.*

I am puzzled by the mathematical treatment of the ozone reactivity. Approximating the calculation of R(O3) using a Taylor series (Supplement A) seems completely unnecessary to me, given that the rate equation has a very simple analytical solution. More importantly, throughout the text the authors report ozone reactivity in terms of Delta(O3), which is not correct. Delta(O3) is the difference between the ozone measured before and after the reactor, from which ozone reactivity (which is in s-1) can be calculated. It is not just a matter of using the wrong unit, it can also cause incorrect results since reactivity depends on the ratio not on the difference of the two ozone measurements, as the authors themselves show with equation S5.

*The referee is right that a simple analytical solution is available. The use of the Taylor series is meant to provide a more elegant formula to calculate ozone reactivity. As $\Delta[O_3]$ is always much smaller than $[O_3]_0$, the condition for applying the Taylor function is always fulfilled (typically $\Delta[O_3]$ is only a few percent of $[O_3]_0$). In this case, the Taylor function leads to the same values as the exact analytical solution. We rephrased the mathematical treatment in the Supplement A to reflect this.*

*In addition, we reviewed our use of the term "ozone reactivity" for $\Delta[O_3]$. We now refer to it as "differential signal" instead where appropriate in the revised manuscript.*

A few comments on the technical side of the instrument.

1) I don't quite understand what the advantage is of using four flasks as a reactor (as opposed to a linear reactor used by other studies). I get it that it makes the system compact and portable, but is there any other advantage with respect, for example, to the mixing of the sample with the ozone reactant or with the residence time? Why four flask instead of 2 or 6 with equivalent total volume?

The design choices of the instrument should be explained, especially if it is claimed that they lead to improvements over other similar instruments.

***We did not intend to claim that the design of the reactor lead to improvements over other similar instruments. The main claim of our manuscript is that the improvement comes from the use of an ozone monitor in differential mode. As highlighted by the reviewer, the main idea was to have a compact and portable system. These flasks were readily available in the CU laboratory and were selected for practical reasons and for their known "inertness and purity towards atmospheric trace gases", as mentioned in the manuscript. The number of flasks (4) was chosen in order to reach a volume of 10L, so that the residence time in the reactor in this configuration is enough to ensure a large enough measurable differential signal. Furthermore, the use of four flasks is a compromise: fewer flasks lead to a "sharper" distribution for the residence time (see section 3.4) compared to using more flasks ("wider" distribution), but the volume would be too small to ensure a clear differential signal. Furthermore, the use of four flasks with dip tubes ensure "efficient purging and replacement of the air volume inside the flasks with minimal mixing" as mentioned in the manuscript. Four flasks also were a very good fit for using up the available space in the insulated Pelican box that was chosen for the reactor storage container.***

2) From figure 3, it seems that ozone is added to the sample before the mixer and then the flow of ozone+sample is split before it enters the 4-flasks reactor. Surely this introduces an error in the determination of ozone reactivity, as BVOC start reacting with ozone in the mixer and the measured "O3 before the reactor" results lower than it actually is. This of course depends on the residence time in the mixer and along the lines that connect it to the reactor, so it may be negligible, but the authors should address this potential issue.

***The volume of the mixer and the tubing leading to the reactor is estimated to be about 15 ml. At a flow rate of 5 L min$^{-1}$ (total prior to sampling before the reactor), the residence time in this small volume is very short (negligible), on the order of < 0.2 s, which is a small (i.e. neglibible) fraction (~ 0.1 %) in relation to the overall residence time of ~ 150 s. We added a sentence to the manuscript to address this.***

3) It is repeatedly stated that the reactor flow in the default configuration is 5 slpm. However, from figures 3 and 5, it looks like the actual flow is 4.4 slpm (4.5 sample + 0.5 ozone - 0.6 to the monitor). On page 22, the reactor flow is declared to be 3.6 slpm. What is the actual reactor flow? If different flows were used for different experiments/measurements, it should be clearly stated and it should be explained why it was necessary to do so.

***Indeed, the instrument has been used in various configurations during testing, which also affected the flow through the reactor and the residence time, for instance when additional monitors (for O₃, and the fast NO monitor) were added after the reactor. A target flow of 4 L min$^{-1}$ was chosen to yield a large enough residence time through the 10 L reactor and a well measureable differential ozone signal. The annotations in Figs. 3 and***

*5 were not accurate and have been updated in the revised manuscript. Figs. 3 and 5 now show the instrument operated in the default configuration with flow through the reactor of 4 l min$^{-1}$. Other occurrences have been corrected in the manuscript. For most experiments and tests, the flow was around 4 l min$^{-1}$ and is reported accordingly, as for instance, on page 22.*

4) I think that the discussion of the detection limit (page 13) is misleading. The sensitivity of TORM is not "slightly higher" than the sensitivity of the Matsumoto (2014) instrument: the difference is about a factor of 2, similar to the difference with the Sommariva et al (2020) instrument. In any case, the actual detection limit of the TORM instrument is of the order of 1e-4 s-1 (page 22), which is higher than both the Matsumoto (2014) and the Sommariva (2020) instruments.

*There seems to be an important misunderstanding. "limit of detection" and "sensitivity" are meant to be two different things and we do not use them interchangeably. Limit of detection is the smallest ozone differential signal that can be quantified with confidence outside of the noise range of the differential ozone signal determination. The LOD was determined as three standard deviation of the delta ozone signal variability during sampling of clean, BVOC-free air. The hourly standard deviation values for delta ozone were 0.075 - 0.096 ppb and are discussed in section 3.1., yielding a delta ozone LOD of 0.23 – 0.29 ppb.*

*Sensitivity (note that from here on, we substitute this term with 'response') defines the ozone differential signal per reactivity of BVOC in the sample flow. An upper threshold of the sensitivity was calculated from the experiment with the limonene standard. The slope of the Figure 9 graph depicts the delta ozone as a function of the ozone reactivity ($R_{O3}$), as determined from the theoretical limonene mole fraction multiplied by its ozone rate constant. The linear regression slope value using the experimental values accounted to 6.2 x 10$^3$ ppb s assuming a 20 ppm mole fraction of the standard. This should be considered a lowest case estimate. If, for instance, the actual mole fraction of the standard was half of its preparation value (10 ppm), the slope value of the regression would account to 1.2 x 10$^4$ ppb s. Figure 9 also depicts the theoretically expected behaviour of delta ozone versus the ozone reactivity (at 20 ppm standard mole fraction) based on the assumption of perfect mixing and the experimental residence time. There is approximately a factor of 2 difference between the two data series. The theoretical regression result (1.4 x 10$^4$ ppb s) would be the response value if this difference is due exclusively to the standard having a lower mole fraction than the one assumed from its preparation.*

*We have added more explanation of these two terms in the text. Further, in order to avoid this confusion, we replaced the term "sensitivity" with "response" where applicable.*

5) After pressure balancing, the authors indicate that the ozone measurement artifact is about 1.7 ppb. Were the data corrected for this artifact? Is the artifact dependent on any ambient parameters (pressure, temperature, humidity)? Why does figure 5B shows 2 valves and figure 5A shows only 1? It would also be good to know whether the valve added to control the pressure can cause any significant loss of ozone.

*In the revised manuscript, we have clarified that the 1.7 ppb value is valid for the specific configuration for which the pressure balancing was performed. Discussing Fig. 6B, we now mention that this value can vary according to the instrument conditioning and configuration.*

*For simplicity, Fig. 5A does not include the setup for the counterflows of the Nafion driers. Figure 5B on the other hand depicts only the plumbing for these counterflows. Therefore, the valve in Fig. 5A (Teflon) is the one used for pressure balancing and the valves in Fig. 5B (stainless steel) are the ones used to make sure that the counterflows in the Nafion driers are 2 to 3 times the sample flow. As mentioned, the valve in Fig. 5A is made of Teflon, so that this added control of the pressure does not cause a significant loss of ozone. As no sample flow goes through the stainless-steel valves in Fig. 5B, they do not affect ozone measurements in any way. The caption of Fig. 5 has been updated to clarify this.*

Section 3.3. Why was it necessary to normalize the reactivity measurements to the air flow and the weight of the branch? A reference to Supplements C and D, and a basic description of the experimental setup for these experiments is missing from the text. It is also not clear what the "blank experiment" was: supplement D mentions a "soil chamber enclosure", which seems to suggest a different type of chamber than the one used for the branch enclosure experiments, but there is not enough explanation. The points and lines in figure 6B are very hard to see and the y-axis labels in figures 6A and 6C are not clear (what is "API" that is subtracted from the 49C measurements?). It would also appear that the Delta(O3) from an empty chamber (figure 6B) is often higher than the measured Delta(O3) (figure 6A) but I guess that cannot be the case, so some explanation should be added to the text. Was the reactivity measured in the empty chamber subtracted from the reactivity measured in the full chamber?

*Normalizing the reactivity measurements to the air flow and the weight of the leaves' biomass allows for the determination of the ozone reactivity of biogenic emission rates and their normalization to the leaf biomass in the experiment. This is meant to allow for comparison with other subsequent studies.*

*We revised thoroughly Fig. 6 to make it easier to read. "API" referred to the ozone monitor model of that was used after the reactor in this specific configuration. This was done in order to compare the two approaches to measure $\Delta[O_3]$: (1) with two monitors and (2) with a differential analyzer. We now refer only to "two monitors" and "differential" $\Delta[O_3]$ for better clarity.*

Section 3.4. What is the purpose of changing the plumbing of the reactor? It only shows that in the changed configuration the residence time is a little longer. In any case, why was the residence time determined using a 4 slpm flow, when the actual reactor flow is 4.4 (or 5, see comment above)? In the end, the authors settle on a 120 seconds residence time, which suggests that the theoretical value calculated at 5 slpm was used. But this does not make sense as the experiment described in this section indicate that the theoretical value is ~30 seconds too long compared to the actual value. In addition a residence time of 167 seconds is mentioned on page 22 and a value of 150 seconds is used in Supplement B. The residence time is a key parameter of the system, and therefore it should be clear what it is. The work in this section should be better explained and the reasoning behind the choice of the final value used for all subsequent analysis should be clearly explained.

***Changing the flow through the reactor in the reverse direction was done to assess (see line 706) the effect of the dip tube on the flow characteristics of the system. The test confirmed that introducing air through the dip tube results in a narrower distribution for the residence time (lines 716-717), which is advantageous for the purpose of the experiment.***

***A set of tests were performed to characterize and improve the experiment. Those at times required a different plumbing as the default configuration, which affected the flow through the reactor. However, the experimental determination of the residence time as described in section 3.4 was performed only with one system configuration with flows of 4 and 6 l min$^{-1}$ through the reactor, with 4 l min$^{-1}$ used in the default configuration as clarified above. There are various ways to define the residence time, as also discussed by Sommariva et al. (2020), and as depicted in Fig. R1 below. It can be (1) the time when the signal starts to increase, (2) the mode of the distribution, i.e. the most frequent residence time, (3) the mean of the distribution, and (4) the theoretical residence time based on reactor volume and flow rate calculation. In the original manuscript, we used the definitions (2) and (4), leading to the inconsistencies pointed out by the referee. In the revised manuscript, we explain that we use the mean of the residence time distribution (3) for ozone reactivity calculations. These times are 132 s and 79 s for the 4 and 6 l min$^{-1}$ flows, respectively. This corresponds to fractions of 0.88 and 0.91 of the theoretical residence time. When the flow through the reactor deviates from 4 or 6 l min$^{-1}$ at which we have experimental data, we applied a factor 0.9 to the theoretical value in order to utilize a residence time that is more in line with the experimentally determined value.***

[Figure]

**Figure R1.** *Response of the fast NO monitor placed after the reactor after injection of NO in the reactor flow for 4 and 6 l min⁻¹ (in blue and orange, respectively).*

Section 3.5. The authors refer to previous studies and earlier experiments on the effect of humidity on ozone measurements: the appropriate references are missing. At line 730, the authors say that an interference can be caused by the addition of water to the sampling flow. It is hard to judge this statement without information on how much water was added, and whether it is comparable to ambient levels and/or to the levels in the enclosures. It is also not clear what is meant with the statement "The bias in the ozone recording lasted significantly longer (10 times) then the residence time". Was the interference significantly larger than the inherent variability of the ozone source? There is not enough detail on these experiments and their description is not clear. I also assume that the reactivity data were corrected for the residual water interference on ozone: supplement G clearly shows that the combination of a Nafion dryer with a differential monitor reduces but does not eliminate the intereference, so it is misleading to state that this setup eliminates the need for correction algorithms (lines 691-692).

*The first reviewer comment relates to the introduction section. We added references to Wilson and Birks (2006) and Spicer et al. (2010) studies there as well.*

*The statement in lines 691-692 was meant to explain that if the determination of the ozone loss in the absence of chemical gas phase reactions (previously referred to as "background", but now for better clarity termed "ozone wall losses" are performed at a similar RH as the ozone reactivity measurements, the Nafion dryers ensure a negligible interference despite changes in RH. This is now clarified in the manuscript. One should also emphasize that during real applications the dynamic range of humidity changes is generally smaller than the wide range that was tested in the lab experiment. Scrubbed ambient air, with ambient RH levels, was always used for zero tests.*

Section 3.6 (laboratory test). I do not understand the point of this section. Figure 9 shows that the theoretical reactivity based on the assumed concentration of limonene is linearly correlated with the measured and modelled reactivity. There are several problems with this: first, the authors do not know exactly the concentrations of limonene being measured, nor they provide an uncertainty

estimate. Second, the modelled reactivity (which model? a model is also mentioned on page 4 and Supplement B but no details are given anywhere) is more than a factor of 2 higher than the measured reactivity and the authors explain the discrepancy by saying that it is "likely" due to the uncertainty in the limonene standard. A factor of 2 would imply that there is a major issue with the limonene standard used. Therefore I am not sure what conclusions could or should be drawn from Figure 9 and the associated discussion.

*It is true that the large uncertainty in the mole fraction of limonene is unfortunate. However, this experiment still serves the purpose for demonstrating the linearity of the system's response to a linear increase in the limonene, despite not knowing the absolute mole fraction. The model mentioned here, on page 4, and in supplement B is a simple box model using reactions of BVOCs with $O_3$ and solved with a Kinetics Pre-Processor. It is meant to estimate the $O_3$ decay in the reactor (i.e., $\Delta[O_3]$). We added a paragraph to clarify this in the methods section.*

*As the limonene standard was prepared in house and had been stored for a relatively long time and it was prepared in a non-specialty treated cylinder, it is not unrealistic to assume that the mole fraction could have decreased by a factor two.*

Section 3.6 (ambient data). Two days of data from a branch enclosure experiment are shown in Figure 10, but the discussion is severely lacking. The authors mention, but do not show, concurrent observations of BVOC: even if they will be the subject of a future paper some data should be shown here, as they can help understand how well the instrument is performing. The authors also mention, but do not show or elaborate, that reactivity and "normalized reactivity" are different by a factor of 3. As I mentioned before, the need for normalization should be justified, it should also be explained why the normalized data are so different, and what does it mean for the interpretation of the results presented here.

*Indeed, the idea of this section was to showcase the data from the application of a branch enclosure experiment. Allocating the observed ozone reactivity to individually identified BVOCs will be presented in a follow-up manuscript that is currently in preparation.*

*The sentence regarding the normalized reactivity was unclear and has been edited in the revised manuscript. We meant to state that there is a variation up to a factor of three for the normalized reactivity for the various tree species investigated.*

MINOR COMMENTS

Figure 8: please do not use "ppt" to indicate "parts-per-thousand". It is normally intended to mean "parts-per-trillion".

*This was a spelling error. We intended to use "ppth. We now use the permille symbol (‰) in the revised manuscript.*

line 578: what "protective film"? Please be more specific.

*The protective film is a polyelefin shrink wrap (buyheatshrink.com).*

line 603: "OH" not "ozone" scrubber.

*We replaced "ozone scrubber" with "OH scavenger".*

---

## Author Comment (AC2)

**Ozone Reactivity Measurement of Biogenic Volatile Organic Compound Emissions**

Detlev Helmig[1,2]*, Alex Guenther[3], Jacques Hueber[1], Ryan Daly[1] , Wei Wang[1] , Jeong-Hoo Park[1], Anssi Liikanen[4], Arnaud P. Praplan[4]

[1]Institute of Arctic and Alpine Research, University of Colorado, Boulder, CO 80309, USA
[2]now at: Boulder Atmosphere Innovation Research LLC, Boulder, CO 80305, USA
[3]University of California Irvine, CA, USA
[4]Atmospheric Research Composition, Finnish Meteorological Institute, 00101 Helsinki, Finland
**\*corresponding author: dh.bouldair@gmail.com**

*Manuscript submitted to*

Atmospheric Measurement Techniques

**Response to Anonymous Referee #2**

Helmig et al. present the development of a total ozone reactivity monitor (TORM) for the direct determination of the ozone reactivity of vegetation emissions. The authors first describe the method and the modification brought to the system to minimize errors and interferences. A commercial UV absorption monitor has been modified to measure directly the difference of ozone before and after the reactor instead of using two monitors. In addition, Nafion dryer membrane tubing have been used before the two inlets of the monitor to reduce the known interference from water vapor of this kind of instrument. The authors then present the different tests they conducted to characterize the instrument including ozone loss on the vessel walls, pressure difference between the two channels of the instrument, evaluation of the modified monitor, estimation of the residence time, assessment and mitigation of humidity effects. The authors finally present some application examples including the measurement of ozone reactivity of test mixtures and samples from vegetation enclosures. On the whole, the characterization tests performed are not well described and appear insufficient to ensure the good quality and reliability of the measurement of ozone reactivity conducted by the instrument. Nevertheless, this manuscript is within the scope of AMT and will be of interest for the atmospheric community. I therefore recommend publication in AMT but after major revision.

***We are grateful for the comments from the referee. We have revised our manuscript to better describe the tests performed and demonstrate the performance of the ozone reactivity measurements. Following are the detailed answers to the comments (in italics).***

**Main comments**:

1) The authors described the setup of the two instruments with four flasks of 2.5L but no explanation is given on this choice. Why four flasks in series and not one or two bigger flasks of the

same volume? This setup does not seem to be optimal and the part where a loss of reactants (ozone or biogenic VOCs) can occur is multiplied.

*The original manuscript states that "The glass flask reactor design was chosen because it was deemed more compact and robust for field deployment applications." (lines 572-574). In addition, they "have been developed and extensively tested for their inertness and purity towards atmospheric trace gases" (lines 576-577), which includes biogenic BVOCs. Because these flasks were available in the CU laboratory, they were selected for the reactor design. These glass flasks have proven to be the most inert material for collection of greenhouse gases. Air samples are collected in these flasks from sites all over the world and then shipped back to the Boulder NOAA lab for analyses, at times six months after sample collection. Pollmann et al. (2008) demonstrated storage of hydrocarbons in these flasks for a period of up to one year. These flasks are among the most inert air sample storage vessels known. BVOCs that reach the reactor through the sampling lines from the branch enclosure are unlikely to be lost to the inert walls of the flasks given they are placed in a heated housing and the relative short residence times. We added this information as motivation to the design of the reactor/choice for the flasks.*

The wall loss is partially explored by the authors in the section 3.1 (system conditioning) where a procedure for passivation of the system with ozone is performed. However, the authors stated that the loss wall was reduced to 1-2 ppb and did no longer show any drifts in the signal (P13, L630-631). It is not clear if this 1-2 ppb loss of ozone is something that remain constant over time after conditioning of the system and that is reproducible from one experiment to another and how is it taken into account in the measurement?

*In the revised manuscript, we now discuss in detail that this value remained constant for a given setup as long as the conditions in the system remain the same.*

To complete this wall loss assessment, estimation of the wall loss for VOCs is also needed, especially for monoterpenes and sesquiterpenes, to determine how it impacts the ozone reactivity measurements?

*We did not perform BVOC recovery experiments in the flow reactor. Most of the surface area of the reactor is borosilicate glass, which is about the most inert wall material for VOCs that is known (please also see our response to the previous comment). Further, the reactor was slightly heated, which further reduces wall losses of heavier VOCs. For instance, we previously demonstrated storage of VOCs to up to 14 carbon atoms over 2+ years in a heated Aculife-treated aluminium cylinder (Helmig et al., 2004).*

*Residence times in the experiment here were much shorter, on the order of minutes, i.e. only approximately $1/10^{7}$ of the longest time that was tested in the referenced experiments. Furthermore, as already stated above, we consider it unlikely that BVOCs*

*that are volatile enough to be purged out of the branch enclosure and reach the reactor through the sampling lines are subsequently lost to the glass walls of the heated flasks.*

2) In section 3.2: Balancing of the ozone monitor inlet pressures, the authors report an ozone differential signal of 1.7 ppb between the pre- and the post-reactor inlet. What is the cause of this difference? How is it taken into account in the measurements? Does it correspond to the 1.7 ppb subtracted from the measurement in the application examples (P 21, line 787-788). If it is the case please clarify. What is also not clear is how often this "background" is measured and does it remain constant over time? What is the procedure applied if differences are observed for this background before and after an enclosure experiment?

*In the original manuscript, "Background" referred to the differential signal in the absence of reactive species (e.g., zero air measurements) and correspond to wall losses of $O_3$ in the reactor. This value can vary at the beginning of the experiment as the system equilibrates and when the flow through the reactor is modified. This is now explained in more detail in the revised manuscript. The ozone wall loss was determined regularly in the field by conducting measurements with scrubbed ambient air and an empty bag. Those measurements were then used as the reference/zero value for vegetation enclosure experiments.*

3) In section 3.5: Evaluation and Mitigation of humidity effects, the authors report a residual ozone reactivity signal response of 0.5 ppb for the differential monitor over a range of relative humidity of 10 to 84% and a residual response six times larger for the two-monitor instrument. This difference in interference is also observed in supplement G. Since both system were sampling through the Nafion tubing, what is the explanation for such difference in the interferences observed by both systems? How the remaining humidity interference observed for the differential monitor is taken into account in the measurement of ozone reactivity?

*The schematic of the instrument was misleading. Nafion dryers were used only for the direct monitoring of the differential signal with one monitor. We did not dry the sample air when two monitors were used. The difference for the two-monitor signal stems from the differences in the response of the two monitors to the change in water vapor in the sample stream. Since two instruments will likely respond different towards humidity changes, there is a high likelihood that the effect on the difference in the signal between the monitors is going to be larger (unless the effects happen to cancel each other out). We have updated the schematic of the instrument in the revised manuscript.*

4) In section 3.6: application examples, the authors compare the ozone difference measured by the TORM and theoretical ozone depletion expected from the reaction of ozone with introduced limonene considering theoretical limonene concentrations, reaction rate constant and theoretical residence time. This comparison resulted in large discrepancy between the theoretical and the measured ozone depletion. The authors explain this discrepancy by the fact that the concentrations of limonene inside the cylinder is uncertain and is expected to have decreased with

time. Furthermore, the authors used the theoretical residence time determined from the volume of the reactor and the flow rate.

First, why using the theoretical residence time since this latter was determined experimentally?

***In the revised manuscript, however, we establish that the peak residence time corresponds to about 80% of the theoretical residence time for the instrument. The paragraph in section 3.6 and Fig. 9 have been updated accordingly.***

Then, this test is very important to perform a quality control and to ensure the reliability and good quality of the measurements performed by the TORM instrument which is not possible with the experiment shown in the paper. I would therefore suggest to perform again this experiment but with certified and known amounts of a BVOC or even better repeat it for several BVOCs (monoterpenes and sesquiterpenes) to check the response of the instrument and compare it to an accurate theoretical ozone depletion. I also suggest to use the residence time determined experimentally for the calculation of the theoretical ozone depletion.

***None of the manuscript authors are currently employed by the University of Colorado and the laboratory and instrument are no longer available for further experimental work. This research, however, is continued at Helsinki University. The proposed experiments are planned to be conducted by the Finnish group in their future TORM research and then be reported in a future publication.***

**Minor comments**:

-P3, line 428: Change "methyl chavicol can be an important emission" for "methyl chavicol can be strongly emitted"

***We changed the sentence according to the referee's suggestion.***

-P3, line 433: Change "BVOC emissions" for "BVOC concentrations"

***We replaced "emissions" with "concentrations".***

-P10, line 578: "Flasks are covered with a protective film".

What is this protective film made of?

***The protective film is a shrink tubing material.***

-P10, line 579-580: "one valve connects to a dip tube that leads to the inside on the opposite side of the flask (Fig. 4)". This is not visible in Fig. 4. Please remove the reference to the figure or use a picture in Fig. 4 where it is visible.

***We rephrased the sentence.***

-P12, line 603: Change "ozone scrubber" for "OH scavenger".

***We changed the sentence according to the referee's suggestion.***

-P16, Figure 6: The results in panel b are hardly visible. Please modify this panel to improve its quality. Please use reasonable significant figures for the linear fit equations in panel C.

*We have updated the figure accordingly to the referee's recommendations.*

-P17, lines 713-715: "Nonetheless, the residence time of ≈120s for the normal plumbing configuration is sufficient to meet the requirements for the ozone reaction experiment"

What do you mean by sufficient? Please clarify and be more specific.

*In this context, it was meant that the residence time should be long enough to allow for $O_3$ to react with the BVOCs sampled and give a large enough differential signal that can be measured with enough precision with the given TORM configuration. We have rephrased this sentence differently in the revised manuscript.*

-P21, line780: Change "reported" for "theoretical".

*This change was implemented as suggested change.*

-P21, Figure 9: Please modify the format of the number of the x axis to scientific notation. Please use reasonable significant figures for the linear fit equations.

*We have updated the figure according to the referee's recommendations.*

-P22, lines 807-808: Change "25 parts per thousand" for "2.5%".

We implemented the change.

-P25, Figure 11: This figure is of poor quality, please modify it to improve its quality.

*We have improved the quality of several figures in the revised manuscript, including Fig. 11.*

**References cited**

Helmig D., Revermann T., and Hall B. (2004) Characterization of a pressurized C5-C16 hydrocarbon gas calibration standard for air analysis. Anal. Chem. 76, 6528-6534.

Pollmann, J., Helmig D., Hueber J., Plass-Duelmer C., and Tans, P. (2008) Sampling, storage, and analysis of C2–C7 non-methane hydrocarbons from the US National Oceanic and Atmospheric Administration Cooperative Air Sampling Network glass flasks. J. Chrom. 11988, 75-87.

---

## Referee Report (RR1)

**Manuscript title**: Ozone reactivity Measurement of Biogenic Volatile Organic Compounds Emissions

Authors: Helmig et al.

https://doi.org/10.5194/amt-2021-354

The authors have addressed almost all of my concerns from the previous version, as well as those of the other reviewer. However, I am still concerned by the comparison between the ozone difference observed by the TORM and theoretical ozone depletion expected from the reaction of ozone with introduced limonene presented in section 3.6. Since this comparison leads to large discrepancy between measured and theoretical ozone depletion probably due to unknown Limonene concentration introduced in the system and since the authors are not in capability of performing again this experiment with a certified standard the outcome of this experiment is very low. If the purpose was to demonstrate the linearity of the TORM response, I would recommend to keep the measurement only and to remove the theoretical value since the comparison brings more confusion than interesting results and to rewrite this section accordingly. The author could also add a sentence to justify the absence of calculation of theoretical value due to uncertain amount of Limonene introduced in the instrument.

---

## Author Response (AR2)

**Manuscript title:**

**Ozone reactivity Measurement of Biogenic Volatile Organic Compounds Emissions**

Authors: Helmig et al.

https://doi.org/10.5194/amt-2021-354

**Answers to the referee comment**

We are grateful for the referees' and editor comments and have addressed them as best as we could in the latest version of the manuscript. Below are our more detailed answers to the referees' concerns. The referees' comments are in italics font and our answers in regular font.

**Referee #1**

*The authors have addressed almost all of my concerns from the previous version, as well as those of the other reviewer. However, I am still concerned by the comparison between the ozone difference observed by the TORM and theoretical ozone depletion expected from the reaction of ozone with introduced limonene presented in section 3.6. Since this comparison leads to large discrepancy between measured and theoretical ozone depletion probably due to unknown Limonene concentration introduced in the system and since the authors are not in capability of performing again this experiment with a certified standard the outcome of this experiment is very low. If the purpose was to demonstrate the linearity of the TORM response, I would recommend to keep the measurement only and to remove the theoretical value since the comparison brings more confusion than interesting results and to rewrite this section accordingly. The author could also add a sentence to justify the absence of calculation of theoretical value due to uncertain amount of Limonene introduced in the instrument.*

As both referees agreed that Fig. 9 is not compelling due to the uncertainty of the limonene concentration, we removed references to the mixing ratio of the test gas from the text as well as the comparison with the modelled values and are now only focusing on the linearity of the system response and use the MFC flow of the standard for the x-axis scale.

**Referee #2**

*The manuscript has significantly improved compared to the initial version, but there are still a couple of problematic points.*

*The main concern I have is about Figure 9 and the associated discussion. I understand that one of the points is to show the linearity of the response. But this is not (or rather not only) what the figure is showing. The x-axis indicates a "theoretical ozone reactivity" in s-1, which strongly implies that is the expected reactivity based on the amount of BVOC sampled. This is highly misleading given that the authors admit that they don't know the exact amount of BVOC used in the experiment (line 482). My suggestion is either to remove Figure 9 entirely (and the corresponding lines 476-499) or rework the figure to use arbitrary units on the x-axis and reword the text (including in the conclusions and abstract)*

*to make clear that this is just to show linearity of response and to remove all references to actual reactvity values.*

As referee #1 expressed the same concern, we have followed the advice from both referees and simplified Fig. 9 as well as the discussion associated with it by removing any reference to specific mixing ratios of the limonene test gas and instead focus on showing the linearity of the response.

*The second issue is with the way the authors seem to use the difference between ozone measurements before /after the reactor (delta) and ozone reactivity. Although the instrument presented here measures O3 reactivity, most of the discussion and the figures are about delta ozone. The two parameters are related, but they are not equivalent and should not be treated as such. Additionally, I think that the way the authors calculate reactivity with the equations in Supplement A is unnecessarily complicated, but I concede this may be a personal preference. In any case, I suggest that the authors calculate the reactivity from delta ozone, and amend the figures, captions and text accordingly because reactivity is the variable that is the focus of the paper and the objective of the instrument.*

We unfortunately do not fully understand the referee's comment. What the instrument measures, physically, is $\Delta[O_3]$. and the total ozone reactivity is derived from this value. For this reason, we discuss in sections 3.2 to 3.5 the effect of pressure, differential measurement, residence time and humidity on $\Delta[O_3]$, including ways to eliminate or mitigate errors during the measurement of this physical parameter. Figure 6 shows $\Delta[O_3]$ as its objective is to show the difference between a two-monitor system and the use of the differential monitor.

The way we calculate total $O_3$ reactivity ($RO_3$) is the same as in Matsumoto (2014), eq. (6) and Sommariva et al. (2020), eq. (4). These are the same calculations as our eq. (S5). The derivations of these equations vary slightly between the publications, but ultimately describe the same calculation. We simply made the derivation explicit in Supplement A. We understand that the reviewer thinks that the approximation that we use to derive eq. (S6) is unnecessary, but we believe that it is an elegant way to calculate $RO_3$. It also highlights that under the operating conditions of TORM ($\frac{\Delta[O_3]}{[O_3]_0} < 0.1$), $RO_3$ is linear to $\Delta[O_3]$. $RO_3$ is derived from equation (S6) for the measured $\Delta[O_3]$, the fixed amount of ozone getting into the reactor ($[O_3]_0$), and the given residence time ($\Delta t$), taking into account wall losses. Because of this linear relationship, any plot with either $\Delta[O_3]$ or $RO_3$ would look the same, besides for the axis scale.

We updated Fig. 10 in section 3.6 on TORM's applications to include $RO_3$ as well. We want the reader to understand, though, that in some cases, especially with emission measurements that can include fast reacting compounds (i.e. with ozone reaction rates similar to the one of NO), the total $O_3$ reactivity derived from TORM measurements with equation (S6) would yield a lower reactivity value than expected from the mixing ratio of the fast reacting compound and its reaction rate with $O_3$ ($k_{O3}$). This is what we wanted Supplement B to illustrate. We are planning to discuss this aspect in the subsequent publication where measured $RO_3$ with TORM and calculated $RO_3$ (from BVOC measurements) will be compared, but we thought that it would be important to be mentioned here as well.

It is true that while Fig. 11 y-axes were labelled "R(O3)", the actual data in the figure were normalized $\Delta[O_3]$. We have updated the figure so that it now shows the normalized measured total $O_3$ reactivity, and the discussion has been slightly altered to reflect this change.

*MINOR POINTS*

*Figure 2: correct "Pluming".*

We have fixed the spelling error in Fig. 2 in the revised manuscript.

*Line 263: please add a number. How much is "negligble"?*

We estimated that the residence time in the tubing is in the order of milliseconds, which is several orders of magnitude smaller than the residence time in the reactor. We have included this information in the text of the revised manuscript.

*Lines 276-279: there is not enough information about the model. Is it assuming only the first step of ozonolysis or does it include the complete oxidation mechanisms? In this case which chemical mechanism was used?*

As Fig. 9 does not use the modelled data anymore, the description of the model was moved from the methods section to Supplement B. There we have added that no secondary chemistry is considered because the typical residence time in the reaction is of a few minutes so that we use the reaction rates at room temperature listed in the legend of Fig. S1. This is now explicitly stated when describing the model.

*Figure 10: is this the same experiment shown in Figure 6? please clarify in captions and text, as appropriate.*

We regret that this was not clear enough. As stated in the text, Fig. 10 data are from field experiments at the University of Michigan Biological Station (UMBS). This information is now also included in the caption of Fig. 10 in the revised manuscript.

*Figure 11: The caption says "delta ozone" and the label on y-axis says "R(O3)". Please correct.*

The reviewer correctly noticed a discrepancy (as discussed above). Despite the y-axis stating "R(O3)", the plotted parameter was a normalized $\Delta[O_3]$. The figure has been updated so that it now shows the measured total $O_3$ reactivity of the emissions (i.e. normalized to the leaf dry weight and the flow through the branch enclosure. These are the measurements from the same field campaign as Fig. 10. This has now been clarified with the addition of the location of the measurements in the figure caption.

*Supplement C: it would be good to show some results of the testing of the differential monitor setup. Was the metering valve used in these tests? If not, why was it not necessary, but it is required in the standard setup?*

The setup presented here is only meant to present how the differential monitor was compared to two independent monitors. This configuration was used in most of the tests performed in this manuscript, except for the ones done with the instrument from the Finnish Meteorological Institute, which use the one-monitor (differential) TORM configuration for the delta ozone determination (but with a second monitor sampling the inflow into the reactor to monitor the absolute value of ozone getting into the reactor, not shown in Fig. 3).

A metering valves was used for all the tests, and it is simply a simplification that it was not displayed in this schematic. Fig. S2 in Supplement C was corrected in the revised manuscript.